# Integrating Spatial and Morphological Characteristics into Melanoma Prognosis: A Computational Approach

**DOI:** 10.3390/cancers16112026

**Published:** 2024-05-27

**Authors:** Chang Bian, Garry Ashton, Megan Grant, Valeria Pavet Rodriguez, Isabel Peset Martin, Anna Maria Tsakiroglou, Martin Cook, Martin Fergie

**Affiliations:** 1The Division of Informatics, Imaging and Data Science, Faculty of Biology, Medicine and Health, The University of Manchester, Manchester M13 9PT, UK; 2Cancer Research UK Manchester Institute, The University of Manchester, Manchester M20 4BX, UK; 3Royal Surrey County Hospital, Guildford GU2 7XX, UK

**Keywords:** melanoma prognostication, cellular morphology, spatial analysis, computational pipeline, deep learning, machine learning

## Abstract

**Simple Summary:**

Spatial characteristics, including cell morphology and spatial distribution patterns, hold potential prognostic value in melanoma. This study builds on existing research by exploring these spatial factors through a computational pipeline, including both univariate and multivariate Cox proportional hazards models, supplemented by rigorous cross-validation techniques. Our findings reveal that spatial and morphological features can potentially enhance the prognostication of melanoma, supporting their integration into standard prognostic models. We also discuss potential limitations and outline directions for future research.

**Abstract:**

In this study, the prognostic value of cellular morphology and spatial configurations in melanoma has been examined, aiming to complement traditional prognostic indicators like mitotic activity and tumor thickness. Through a computational pipeline using machine learning and deep learning methods, we quantified nuclei sizes within different spatial regions and analyzed their prognostic significance using univariate and multivariate Cox models. Nuclei sizes in the invasive band demonstrated a significant hazard ratio (HR) of 1.1 (95% CI: 1.03, 1.18). Similarly, the nuclei sizes of tumor cells and Ki67 S100 co-positive cells in the invasive band achieved HRs of 1.07 (95% CI: 1.02, 1.13) and 1.09 (95% CI: 1.04, 1.16), respectively. Our findings reveal that nuclei sizes, particularly in the invasive band, are potentially prognostic factors. Correlation analyses further demonstrated a meaningful relationship between cellular morphology and tumor progression, notably showing that nuclei size within the invasive band correlates substantially with tumor thickness. These results suggest the potential of integrating spatial and morphological analyses into melanoma prognostication.

## 1. Introduction

Melanoma, one of the most lethal forms of skin cancer, presents a significant public health challenge due to its rapid progression and potential for metastasis [1,2,3,4,5]. Traditional prognostic factors, including tumor thickness (Breslow thickness) and mitotic rate, have been pivotal in assessing patient outcomes [6,7,8,9]. Despite this, there is an increasing incidence of thin melanomas with earlier detection and increased public awareness, but even in the thicker tumors, there is considerable variation in outcome [10,11,12]. Therefore, it is important to determine other important prognostic indicators.

Recent studies have revealed that cell morphology reflects certain molecular signatures [13], as well as a previous study of naevoid melanomas in which a subtype was noted to have a better prognosis when it showed features simulating maturation, which often includes distinct changes in cell size and distribution of cells [14]. In addition, research has investigated the prognostic implications of cell morphology in breast cancer [15], and evidence has also highlighted the potential of nuclei size in survival analysis [16,17,18,19,20]. Studies have also proved the potential of spatial proximity between T and PD-L1 expressing cells as prognostic biomarkers [21]. Recent advancements in computational analysis [22,23,24,25,26,27,28,29] offer new methods to explore these dimensions, potentially uncovering novel prognostic indicators [30,31]. Building on these insights, our study specifically aims to delve deeper into the implications of these morphological traits, specifically evaluating whether variations in cell nuclei size, proliferation rates, and the spatial distribution of different cell groups within the tumor microenvironment could offer additional prognostic value for melanoma.

This study introduces a computational approach to integrate spatial and morphological characteristics of melanoma tumors into prognostic evaluations. To supplement the conventional prognostic factors, we implemented a detailed analysis of tumor morphology, specifically the spatial distribution of different cell subtypes and size of cell nuclei, which could provide additional insights into patient prognosis. To this end, we collected a melanoma cohort of 26 patients and conducted multiplexed immunohistochemistry (mIHC) staining on the samples to acquire visualization of the cell distribution within the tumor microenvironment (TME). One sample of the mIHC staining is shown in Figure 1. Furthermore, we developed a computational pipeline to quantify these characteristics across different tumor spatial bands, focusing on their prognostic significance.

## 2. Materials and Methods

This study was structured around a robust computational pipeline designed to analyze spatial and morphological features of melanoma tissue samples. The primary objective was to quantify the prognostic significance of cellular characteristics across different tumor bands and to integrate these insights with established prognostic factors. The overall schematic pipeline is shown in Figure 2.

### 2.1. Dataset Collection

This study includes data from 26 melanoma patients, selected for having a Breslow thickness greater than 1.2 mm and being older than 16 years of age. The study protocol was approved by the St. Georges Hospital Ethics Committee. A detailed Flowchart demonstrating the quality control process is shown in Figure 3.

A 3-plex immunofluorescent staining was performed on the Leica Bond Rx using the Open Research 2 Kit. In the immunohistochemistry protocol, specific antibodies were used to detect target antigens. The Ki67 was targeted using Agilent M7240 antibody, diluted to 1/200 (0.23 μg/mL), and incubated for 30 min to ensure adequate binding. Subsequently, the S100 protein was detected using Leica Flex GA504 antibody, which was diluted 1:4 with Bond Antibody Diluent and also incubated for 30 min. For nuclear staining, DAPI (ThermoFisher, Waltham, MA, USA) was employed. The slides were incubated with DAPI for 15 min. Stained samples were scanned at 20×. magnification using Olympus VS120 scanner to analyse tumor behaviour and microenvironment interactions concisely.

### 2.2. Cell Detection and Measuring

In this study, cell detection and quantification were performed using the open-source software QuPath (version: 0.3.2) [32,33,34,35] and StarDist2D algorithm [36,37,38], a deep learning framework optimized for the segmentation of cell nuclei. The detection threshold was set at 0.5 to balance sensitivity and specificity, ensuring that only cells with a probability score above this threshold were considered for further analysis. The DAPI channel was selected for nucleus detection, leveraging its high contrast for nuclear material. Image normalization was implemented by adjusting intensities to span from the 1st to the 99th percentiles, enhancing image quality. This approach adjusted for variations in intensity across images and minimized the influence of outliers, leading to more consistent and reliable analysis. A pixel size of 0.5 μm was specified to maintain resolution fidelity.

Subsequently, we systematically quantified cellular and nuclear morphology alongside fluorescence intensity across multiple channels. Morphometric parameters, including area, length, circularity, solidity, and diameters, were measured to characterize structural variations within cells and nuclei. Concurrently, intensity metrics—mean, median, minimum, maximum, and standard deviation—were calculated for each cell compartment (nucleus, cytoplasm, and membrane) and across DAPI, Ki67, and S100 channels. The mean values of each compartment of each channel are used for the subsequent thresholding process.

### 2.3. Cell Thresholding

After acquiring the cellular intensity information, an adaptive thresholding approach is applied on each slide to identify the positive and negative cells of each marker. We adopted a 2-phase thresholding approach to distinct positive cells. Specifically, for each slide, the thresholding process contains 2 steps:

The first step of our thresholding approach applies the Otsu method [39] to the expression distribution. The essence of the Otsu method is to define the threshold that maximizes the separability of the foreground and background distributions by scanning all possible threshold values across the intensity distribution. It achieves this by finding the threshold $t^{*}$ that maximizes the between-class variance, $\sigma_{B}^{2}\left(t\right)$, which is defined as
(1)$$\sigma_{B}^{2}\left(t\right)=\omega_{0}\left(t\right)\cdot \omega_{1}\left(t\right)\cdot {\left[\mu_{0}\left(t\right)-\mu_{1}\left(t\right)\right]}^{2}$$
where $\omega_{0}\left(t\right)$ and $\omega_{1}\left(t\right)$ are the probabilities of the two classes separated by threshold *t*, and $\mu_{0}\left(t\right)$ and $\mu_{1}\left(t\right)$ are the means of these classes. The method seeks the threshold $t^{*}$ that maximizes this variance:(2)$$t^{*}=\arg\max_{t}\sigma_{B}^{2}\left(t\right)$$

This step divides the marker expression distribution into two components: $T_{0}$ for the background, containing primarily noise, and $T_{1}$ for the foreground. Only $T_{1}$ progresses to the next step for refined thresholding.

The second step of the thresholding process involves a Bayesian Gaussian mixture model (BGMM)-based thresholding method. Previous research has demonstrated the effectiveness of the Gaussian mixture model (GMM) in thresholding uneven grayscale images [40]. In our study, to better cater to the nature of multiplex immunohistochemistry (mIHC) image data, we utilized a similar approach with BGMM instead of GMM to achieve greater flexibility. Specifically, the logarithm of the $T_{1}$ component is modeled under a 2-component BGMM approach using the Expectation–Maximization (EM) algorithm [41]. In our study, we set the max iteration number as 3000 based on empirical experiments.

The BGMM represents the $T_{1}$ distribution as a weighted sum of its two Gaussian components:(3)$$p\left(x\right)=\pi_{1}N\left(x|\mu_{1},\sigma_{1}\right)+\pi_{2}N\left(x|\mu_{2},\sigma_{2}\right)$$
where p(x) is the probability density function of the data, $\pi_{1}$ and $\pi_{2}$ are the mixture weights of the two components, and N(x|μ,σ) represents the Gaussian distribution defined as in Equation (Equation 4).
(4)$$N\left(x|\mu_{k},\Sigma_{k}\right)=\frac{1}{\sqrt{{\left(2\pi \right)}^{d}\left|\Sigma_{k}\right|}}e^{\left(-\frac{1}{2}{\left(x-\mu_{k}\right)}^{T}\Sigma_{k}^{-1}\left(x-\mu_{k}\right)\right)}$$

The final threshold is defined at the intersection of the two compartments, where
(5)$$\pi_{1}N\left(x|\mu_{1},\sigma_{1}\right)=\pi_{2}N\left(x|\mu_{2},\sigma_{2}\right)$$

Finally, the compartment with the higher mean (μ) is determined as the final positive cell component. One sample illustration is shown in Figure 4

### 2.4. Spatial Analysis

In our spatial analysis, we employed Principal Component Analysis (PCA) [42] to identify the overall orientation of the tumor structure from which we can define three distinct bands within melanoma tissue. The epidermis and melanoma area were manually annotated with the area contiguously S100+ identifying the main tumor area. From this, the invasive margin was automatically identified in the annotation as the portion of the melanoma opposite the epidermis layer. A distance transform was used to create segement the melanoma region into 3 distinct bands: the Superficial Band at the outer edge, the Middle Band, and the Invasive Band at the invasive margin.

PCA was utilized to calculate the estimated thickness of the tumor. By identifying the principal axes of the tumor mass, we were able to measure the longest distance from the epidermis to the deepest part of the tumor along the vertical plane of the tumor. This measurement was then used to divide the tumor thickness proportionally, assigning the innermost third to the Invasive Band, the central third to the Middle Band, and the outer third to the Superficial Band.

Following the demarcation of these bands, S100+ and Ki67+ cells were quantified, their distances to the tumor margin were calculated, and their positions were mapped to the respective zones. The visualization component of the analysis provided plots that integrated the tumor, epidermal, and margin annotations with the spatial distribution of these immunohistochemically stained cells shown in Figure 5.

Aggregate cell counts and cell sizes for each band were determined, allowing us to analyze cell density and morphology variations and distributions across the tumor’s spatial gradient. This stratification provided insights into the organization and potential behavior of melanoma cells relative to their position within the tumor, giving us a clearer understanding of the spatial dynamics in melanoma progression. The acquired spatial data were subsequently integrated into the survival analysis, providing critical insights into the correlation between cell distribution within the tumor bands and patient outcomes, including relapse rates and overall survival.

### 2.5. Survival Analysis

In the Survival Analysis section, we utilize univariable and multivariable Cox proportional hazards models [43] to examine the factors. Hazard ratios (HRs) and 95% confidence intervals (CIs) were estimated from Cox proportional hazards models. Our focus was on evaluating the prognostic value of spatial and morphological variables, establishing their significance in predicting relapse.

The analysis includes the set of risk factors in Table 1:

The term `colocalized’ refers to cells exhibiting dual positivity for S100 and Ki67, markers indicating not only the presence of active tumor cells but also their proliferation. The `Colocalized-to-Total Cell Ratio’ measures the proportion of such colocalized cells against the total cell count within the tumor, while the `Colocalized-to-Tumor Cell Ratio’ assesses them relative to the total tumor cell count. In this study, nuclei-size-related factors are derived from the mean values of these metrics.

To determine the accuracy of these risk factors as predictors, we calculated concordance indices with confidence intervals using bootstrap methods. Additionally, we explored correlations among risk factors to understand their interrelationships in melanoma.

## 3. Results

### 3.1. Clinical Characteristics

The inclusion criteria for the prognosis analysis is the melanoma thickness above 1.2mm, and we ultimately included 26 patients. The 1- and 3-year OS rates were 91.3% and 60.3%. Table 2 summarizes all patient characteristics.

### 3.2. Risk Prediction

A univariate regression analysis was conducted to examine the influence of various factors on melanoma-specific survival across the study cohorts, as detailed in Table 3.

Building on the detailed univariate Cox proportional hazards model results presented in Table 3, several key insights emerge regarding the influence of various factors on melanoma-specific survival. Notably, while factors such as Mitosis rate, Thickness, Estimated Thickness, and Age provide foundational understanding, it is the size-related variables that demand closer attention for their impact on survival outcomes.

The analysis reveals that the mean cell nuclei size, especially within the invasive band, alongside mean tumor cell nuclei size and mean colocalized cell nuclei size across different bands (Invasive, Middle, Superficial), are significant predictors of survival. Specifically, the nuclei size in the Invasive Band shows a significant HR of 1.10 (95% CI: 1.03, 1.18), and the nuclei sizes of tumor cells and colocalized cells in the Invasive Band show HRs of 1.07 (95% CI: 1.02, 1.13) and 1.09 (95% CI: 1.04, 1.16), respectively. These findings underscore the importance of cellular and tumor microenvironment characteristics in determining melanoma progression. Specifically, the Hazard Ratios (HRs) for mean nuclei sizes indicate a clear trend: larger nuclei sizes are associated with a higher risk of adverse outcomes, highlighting the prognostic value of nuclear morphology in melanoma. The consistent association between size-related features and survival suggests that these metrics could potentially inform more tailored therapeutic strategies.

Concordance, along with 95% confidence intervals for each factor analyzed using univariate models, is shown in Figure 6. Notably, variables related to nuclei sizes across the different bands show variable predictive capabilities for melanoma-specific survival. Among these, the nuclei sizes of the Invasive Band stand out with higher concordances, highlighting its potential as a significant prognostic marker. Details of the nuclei sizes of different bands and cell categories are shown in Figure 7.

A multivariable Cox model was used to correct each for the spatial features with respect to the existing clinical risk factors available, including Mitosis and Thickness. Spatial features assessed included the sizes of different cells (all cells, tumor cells and colocalized cells) within the Invasive band. Each Cox model incorporated one of the nuclei-related factors in conjunction with the two traditional factors to assess their combined prognostic significance. This structured analysis will allow us to compare the predictive power of models integrating nuclei morphology information. The three-variable Cox model results are shown in Table 4.

The Concordance Index (C-Index) values for the four multivariate models, distinguished by their variables, are as follows: The base model incorporates two traditional variables (Mitosis and Thickness (mm)). The first three-variate model, incorporating invasive band nuclei size plus two traditional variables (Mitosis and Thickness (mm)), exhibits a C-Index of 0.78 with a confidence interval of (0.62, 0.91). The second three-variate model, focusing on invasive band tumor cell nuclei size along with the same two traditional variables, achieves a C-Index of 0.79 with a confidence interval of (0.65, 0.92). The third three-variate model, which includes invasive band colocalized cell nuclei size in addition to the two traditional variables, also presents a C-Index of 0.79, with a confidence interval of (0.64, 0.93). These findings demonstrate that although the spatial features are prognostic on their own, they are not independently prognostic when combined with the well-established risk factors in this cohort.

Additionally, considering the limited cohort size of this study, to address potential overfitting, which is a concern in studies with small samples and multiple covariates, we implemented a Leave-One-Out Cross-Validation (LOOCV) approach. This method allowed us to assess the stability and reliability of our findings across the dataset and validate whether the results have potential overfitting problems. Specifically, each case was used as a validation set once, with the model being trained on the remaining participants. It reveals how the model performs on unseen data in a repeated and exhaustive manner across the dataset. Furthermore, we calculated the 95% confidence interval on the LOOCV results. This result, detailed in Table 4, demonstrates that the confidence interval for the LOOCV C-Index closely aligns with the confidence interval for the overall C-Index, indicating that our model’s predictive performance is reliable and not overly optimistic due to overfitting.

### 3.3. Factor Correlation

To further reveal the relationship between the risk factors, we quantified the correlation coefficient between the cell nuclei-size-related factors and Mitosis and Thickness (mm). Correlation heatmaps are shown in Figure 8a–c. The three correlation heatmaps illustrate the interrelationships between various morphological characteristics of melanoma and traditional prognostic factors.

The correlation heatmaps reveal a notable finding: beyond the strong intercorrelation of nuclei sizes across different tumor bands, there is a significant correlation between the nuclei sizes in the Invasive Band and the Thickness (mm) of the tumor. This suggests that larger nuclei sizes in the most aggressive area of the tumor are associated with increased tumor thickness, linking microlevel cellular morphology with macrolevel tumor growth. This connection highlights the potential for cellular characteristics within the invasive band to serve as indicators of overall tumor progression, which could be an important consideration for both prognosis and treatment planning. However, it is worth mentioning the correlations identified in this study do not imply causation. We recognize that these associations could be influenced by other factors not explored in this analysis, and as such, they warrant further investigation.

## 4. Discussion

Recent advancements in digital pathology have revolutionized the field, shifting from traditional microscopy to comprehensive digital platforms that enable enhanced visualization, analysis, and the management of histological images [44,45]. This transformation is driven by innovations in high-throughput imaging technologies and machine learning algorithms that facilitate the rapid and accurate analysis of pathology slides [46,47]. Building on the advancements in digital pathology, the integration of spatial information has also led to progress in the field [16,17,18,19,20]. Based on these advances, and taking into account recent findings of melanoma studies [13,14], we aim to establish a computational framework which can integrate morphological and spatial information into melanoma prognostic studies.

This paper presents a computational pipeline designed to quantify the spatial and morphological features of melanoma tissues and evaluate their prognostic value. Specifically, we focused on nuclei sizes and spatial distribution of different cell subtypes. Through this methodology, both univariate and multivariate Cox models were employed to ascertain the prognostic significance of various factors. The univariate analysis particularly highlighted the prognostic importance of nuclei sizes within different spatial bands, with the nuclei size in the invasive band emerging as a factor with a notably high hazard ratio. Such consistency was observed across metrics for all cell nuclei sizes, tumor cell nuclei sizes, and colocalized (KI67+ S100+) cell nuclei sizes, suggesting that spatial analysis could be a valuable tool in melanoma prognosis. The analysis underscores the critical role of cellular morphology in predicting relapse. Moreover, the multivariate Cox models demonstrated an improved mean concordance index upon incorporating nuclei size data, indicating a more accurate prognostic performance when including detailed morphological information. These findings demonstrate the potential of integrating spatial information in prognostic studies of melanoma.

Additionally, correlation analyses were performed to examine the interaction between different factors. The notable correlation between the invasive band’s nuclei size of different cell categories and tumor thickness suggests a link between microlevel cellular changes and overall tumor progression, underscoring the profound impact of cellular architecture on disease dynamics.

In this study, we explore the prognostic value of cell morphology and spatial distribution of different cell subtypes to offer new perspectives for melanoma analysis. Results show that the mean cell nuclei size, especially within the invasive band, alongside mean tumor cell nuclei size and the mean nuclei size of KI-67 and S100 colocalized cells across different bands (Invasive, Middle, Superficial), are significant predictors of survival. The multivariant Cox model also showed that the integration of nuclei size measurements from cells within the Invasive Band has led to an improvement in the concordance of our models compared with the results from univariate analyses. Furthermore, a factor correlation study was conducted to reveal that there is a significant correlation between the nuclei sizes in the Invasive Band and the Thickness (mm) of the tumor. This work has demonstrated the potential of nuclear morphology and spatial information in melanoma prognosis analysis.

In summary, this study highlights the importance of integrating spatial and morphological data with traditional prognostic factors in melanoma analysis. However, we acknowledge the limitation of the cohort size, as this study is designed around a prespecified hypothesis based on histological expertise. As a discovery-oriented investigation, our primary goal was to identify potential patterns and hypotheses for future, more extensive studies. Our findings confirm the prognostic value of nuclei sizes, particularly in the invasive band, and show a clear correlation between cellular morphology and tumor progression. These insights demonstrate the potential for a morphology and spatial-informed approach, which could contribute to the development of prognostic studies of melanoma. In this study, we focused primarily on the histological characteristics of tumors. It is important to note that we did not account for potentially influential variables such as treatment modalities and tumor stage. The exclusion of these factors was primarily due to the initial scope and design of our research but may limit the generalizability and statistical power of our findings. Looking ahead, we plan to expand the scope of our analysis to include a larger cohort, which will allow for the inclusion of a broader range of variables. This next phase of research is designed to investigate the interplay between histological features and immune cell dynamics. By doing so, we aim to provide a more comprehensive understanding of their combined impact on prognosis. Such expansions are expected not only to validate our preliminary findings but also to enhance their statistical significance, offering more robust insights that could inform clinical strategies. 

## Figures and Tables

**Figure 1 cancers-16-02026-f001:**
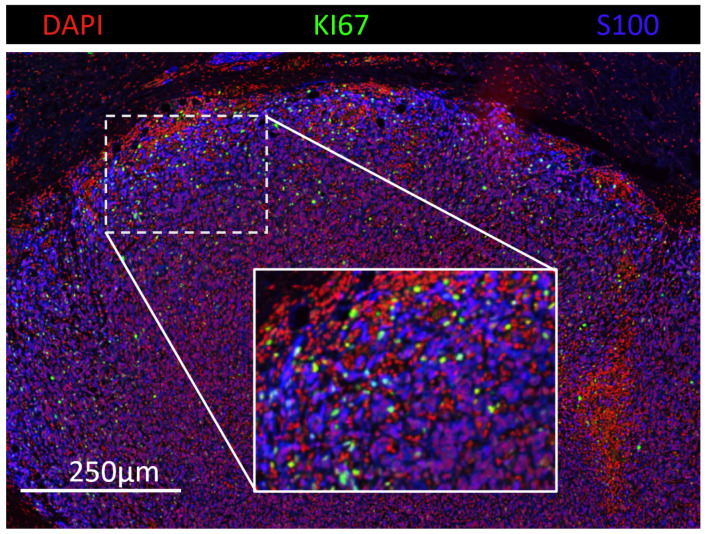
Multiplex Immunohistochemistry (mIHC) Visualization from Our Cohort. The image showcases a melanoma tissue section stained with DAPI (red) for nuclei, Ki67 (green) indicating proliferating cells, and S100 (blue) highlighting melanoma cells. The inset provides a magnified view of the indicated region, demonstrating the cellular detail captured by mIHC.

**Figure 2 cancers-16-02026-f002:**
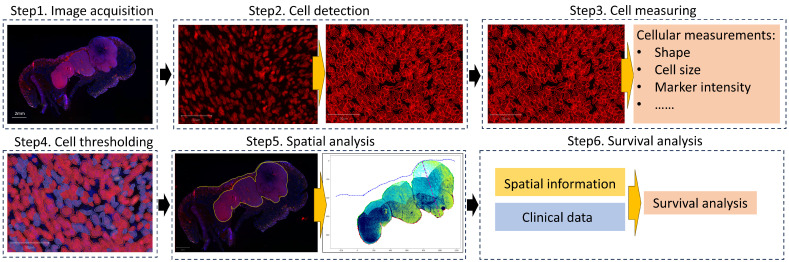
Overview of the computational analysis pipeline for melanoma prognosis: Step1. Acquire the mIHC image slides and annotation of the melanoma site and epidermis. Step2. Detect cells within the melanoma region. Step3. Measure cellular features. Step4. Threshold cells based on the marker intensity to distinguish positive cell counts of different markers. Step5. Quantify spatial and morphological features within different tumor bands. Step6. Conduct survival analysis based on spatial, morphological, and clinical information.

**Figure 3 cancers-16-02026-f003:**
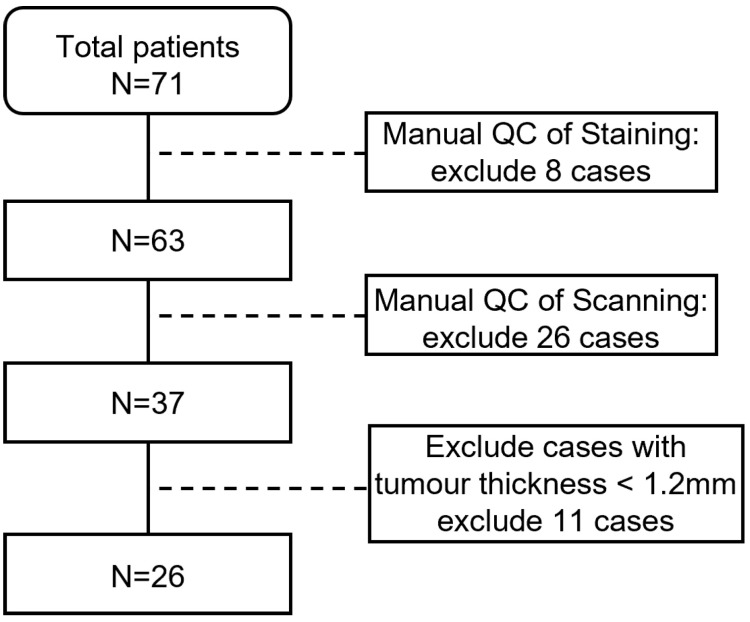
The quality control process of the dataset.

**Figure 4 cancers-16-02026-f004:**
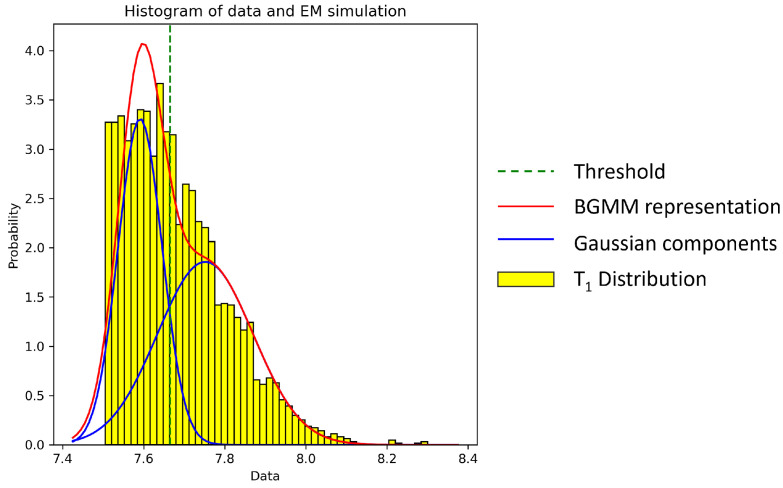
Histogram of the marker expression data overlaid with the BGMM thresholding process, as determined by the EM simulation. The yellow bars represent the $T_{1}$ distribution, the red curve indicates the total fit of the BGMM, and the blue curve illustrates the individual Gaussian components. The intersection of these components, marked by the dashed green line, defines the optimal threshold for segmenting positive cells from negative ones.

**Figure 5 cancers-16-02026-f005:**
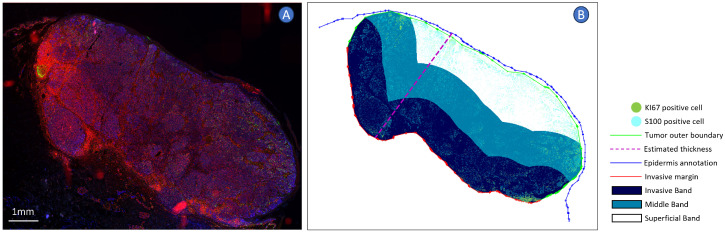
Sample visualization of the spatial analysis process. Panel (**A**) displays a fluorescence microscopy image of the melanoma sample, highlighting S100+ (blue) and Ki67+ (green), and DAPI (red). Panel (**B**) shows a schematic of the spatial analysis results: the melanoma region is segmented into three distinct spatial bands: the Superficial Band (white), Middle Band (blue), and invasive band (dark blue). The thickness of each band is derived from PCA-based tumor thickness measurements, indicated by the dashed magenta line.

**Figure 6 cancers-16-02026-f006:**
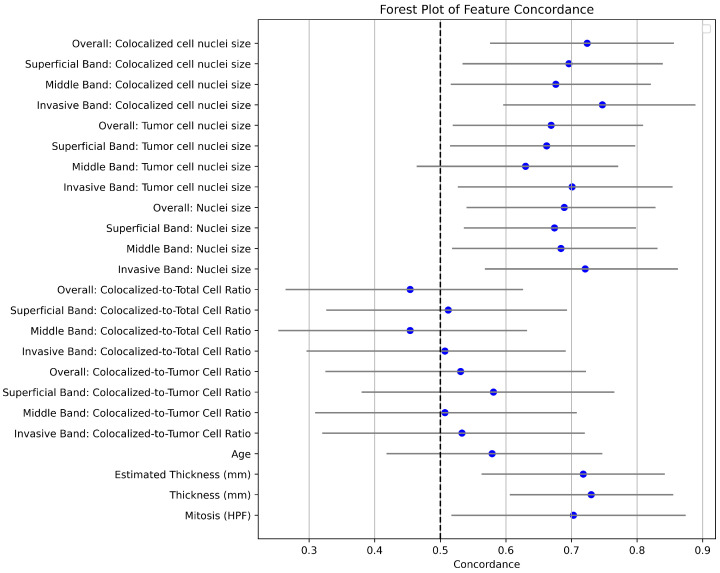
Forest plot of concordance indices for prognostic factors in melanoma. This plot illustrates the concordance indices and their 95% confidence intervals for various prognostic factors, including cell density and nuclei sizes across different tumor bands, as well as traditional factors such as age, mitosis rate, and tumor thickness.

**Figure 7 cancers-16-02026-f007:**
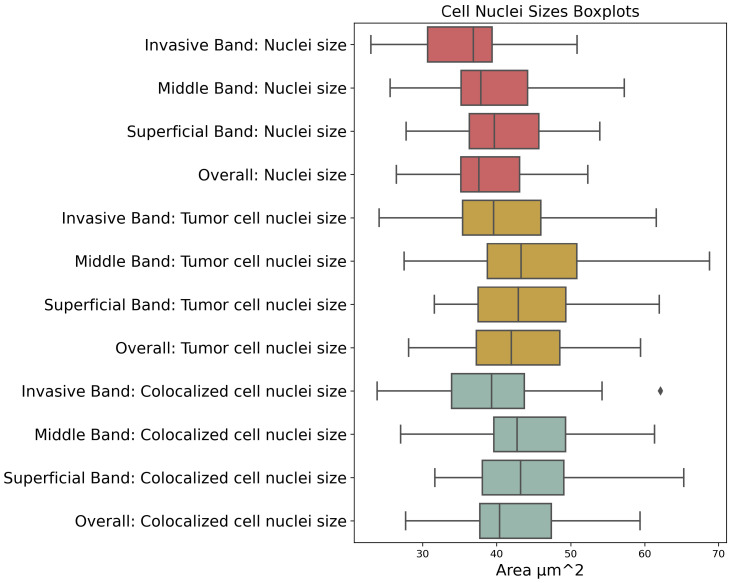
Boxplots of cell nuclei sizes across different tumor bands. The boxplots compare the distribution of cell nuclei sizes within the invasive, middle, and superficial bands of melanoma tumors, including mean cell nuclei size, mean tumor cell nuclei size, and mean colocalized (Ki67+ S100+) cell nuclei size.

**Figure 8 cancers-16-02026-f008:**
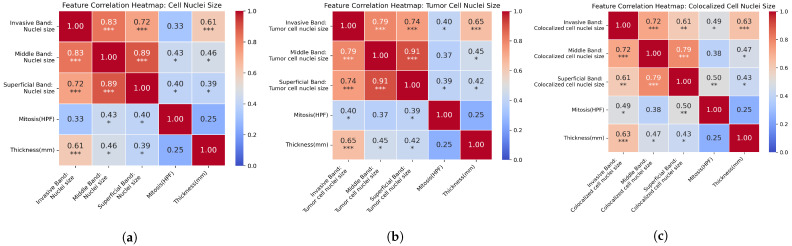
Composite correlation heatmaps of cell, tumor cell, and colocalized cell nuclei sizes with traditional prognostic factors in melanoma. Each subfigure represents the pairwise correlation coefficients between nuclei sizes in different tumor bands and standard prognostic measures, such as mitosis rate and tumor thickness, across three distinct cellular categorizations: (**a**) cell nuclei size, (**b**) tumor cell nuclei size, (**c**) colocalized cell nuclei size. (Asterisks denote levels of statistical significance: * *p* < 0.05, ** *p* < 0.01, *** *p* < 0.001).

**Table 1 cancers-16-02026-t001:** Summary of prognostic factors and their applicability across tumor bands.

Prognostic Factor	Band
Mitosis (HPF)	N/A
Thickness (mm)	N/A
Estimated Thickness (mm)	N/A
Age	N/A
Colocalized-to-Total Cell Ratio	Invasive, Middle, Superficial, Overall
Colocalized-to-Tumor Cell Ratio	Invasive, Middle, Superficial, Overall
Nuclei Size	Invasive, Middle, Superficial, Overall
Tumor Cell Nuclei Size	Invasive, Middle, Superficial, Overall
Colocalized Cell Nuclei Size	Invasive, Middle, Superficial, Overall

Abbreviations: N/A, Not Applicable.

**Table 2 cancers-16-02026-t002:** Clinical Characteristics of the Cohort.

Characteristic	Value	No.	%
Median Age, years	62.5		
Age Range, years	24–78		
Gender	Male	20	76.9
	Female	6	23.1
OS: event	True	16	61.5
	False	10	38.5
Median OS: months	35.5		
PFS: event	True	18	69.2
	False	8	30.8
Median PFS months	27.5		

**Table 3 cancers-16-02026-t003:** Univariable cox proportional hazards model results.

Variables	HR	95% CI		*p* Value
		**Lower**	**Upper**	
Mitosis (HPF)	1.15	1.06	1.24	<0.005
Thickness (mm)	1.70	1.23	2.35	<0.005
Estimated Thickness (mm)	6.01	1.59	22.67	0.01
Age	1.01	0.98	1.05	0.41
Invasive Band: Colocalized Cell Density	0.90	0.45	1.82	0.77
Middle Band: Colocalized Cell Density	0.88	0.51	1.52	0.64
Superficial Band: Colocalized Cell Density	0.84	0.53	1.32	0.44
Overall: Colocalized Cell Density	0.71	0.03	16.63	0.83
Invasive Band: Cell Density	0.82	0.47	1.42	0.48
Middle Band: Cell Density	0.85	0.48	1.51	0.59
Superficial Band: Cell Density	0.99	0.66	1.49	0.98
Overall: Cell Density	0.78	0.40	1.52	0.46
Invasive Band: nuclei size	1.10	1.03	1.18	<0.005
Middle Band: nuclei size	1.08	1.01	1.15	0.02
Superficial Band: nuclei size	1.10	1.02	1.19	0.01
Overall: nuclei size	1.11	1.03	1.20	0.01
Invasive Band: tumor cell nuclei size	1.07	1.02	1.13	<0.005
Middle Band: tumor cell nuclei size	1.05	1.00	1.10	0.04
Superficial Band: tumor cell nuclei size	1.07	1.01	1.13	0.01
Overall: tumor cell nuclei size	1.08	1.02	1.14	0.01
Invasive Band: colocalized cell nuclei size	1.09	1.04	1.16	<0.005
Middle Band: colocalized cell nuclei size	1.08	1.01	1.14	0.01
Superficial Band: colocalized cell nuclei size	1.07	1.02	1.12	0.01
Overall: colocalized cell nuclei size	1.11	1.04	1.18	<0.005

**Table 4 cancers-16-02026-t004:** Multivariate cox model analysis of different nuclei sizes with C-Index.

Variables	HR	95% CI		*p* Value	C-Index	LOOCV C-Index
		**Lower**	**Upper**			
Base model						
Mitosis (HPF)	1.18	1.07	1.29	<0.005		
Thickness (mm)	2.01	1.31	3.07	<0.005	0.77 (0.61, 0.91)	0.78 (0.61, 0.93)
Nuclei Size						
Invasive Band: Nuclei size	1.02	0.92	1.12	0.75		
Mitosis (HPF)	1.17	1.06	1.29	<0.005	0.78 (0.62, 0.91)	0.78 (0.62, 0.91)
Thickness (mm)	1.90	1.11	3.25	0.02		
Tumor Cell Nuclei Size						
Invasive Band: Tumor Cell Nuclei size	1.00	0.93	1.08	0.92		
Mitosis (HPF)	1.18	1.06	1.30	<0.005	0.79 (0.65, 0.92)	0.78 (0.61, 0.92)
Thickness (mm)	1.97	1.16	3.37	0.01		
Colocalized Cell Nuclei Size						
Invasive Band: Colocalized Cell Nuclei size	1.03	0.95	1.11	0.54		
Mitosis (HPF)	1.17	1.05	1.29	<0.005	0.79 (0.64, 0.93)	0.78 (0.61, 0.93)
Thickness (mm)	1.86	1.13	3.05	0.01		

Abbreviations: HR, Hazard Ratio; CI, Confidence Interval; LOOCV: Leave-One-Out Cross Validation.

## Data Availability

Data supporting the findings of this study are available from the author upon reasonable request.

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
