# Peer review of "Integrating Spatial and Morphological Characteristics into Melanoma Prognosis: A Computational Approach"

_cancers, 2024, doi:10.3390/cancers16112026_

Round 1

Reviewer 1 Report

Comments and Suggestions for Authors

Utilizing a computational methodology, the manuscript conducts an exhaustive examination of the prognostic value of cellular morphology and spatial configurations in melanoma. The following are some evaluations and inquiries about the methodology and findings:

The cohort of 26 melanoma patients included in the study is comparatively small. To ensure that the findings can be generalized, how do you intend to resolve the sample size issue in future research despite the promising methodology and results?

The immunohistochemistry protocol appears to be described in detail. Nevertheless, did any obstacles or restrictions manifest themselves throughout the staining procedure that might compromise the precision or replicability of the findings?

The efficacy of QuPath and StarDist2D in detecting and quantifying cells seems to be uncompromised. Nevertheless, what steps were implemented to verify the precision of these techniques, specifically concerning locating nuclei in intricate tissue structures?

The described two-phase thresholding strategy appears exhaustive. Nevertheless, was there any difficulty in the research regarding the precise differentiation of positive and negative cells, particularly in areas characterized by dense cell populations or overlapping staining?

Principal Component Analysis (PCA) is an innovative technique to identify tumor bands and quantify tumor thickness. Nevertheless, this method's sensitivity to variations in tumor morphology begs the question: were alternative spatial analysis techniques taken into account?

In the survival analysis, the Cox proportional hazards models, both univariable and multivariable, offer significant insights regarding the prognostic importance of different factors. Nevertheless, the determination of thresholds for defining significant hazard ratios and any necessary adjustments for multiple comparisons remain unknown.

Examining the correlation between the diameters of nuclei and conventional prognostic factors is enlightening. Nevertheless, it is worth noting that the correlation analysis may have overlooked potential confounding variables or factors that could impact the observed relationships.

It is appropriate to assess the predictive capability of multivariate models using the C-Index. What methodology is employed to compute the confidence intervals for the C-Index, and to what extent do these estimates possess robustness?

A comprehensive overview of the findings and their implications is presented in the discussion. Nonetheless, how were any possible limitations or biases that could have impacted the interpretation of the results accounted for in the study?

Regarding melanoma prognosis, the study emphasizes the potential of integrating spatial and morphological data. What are the forthcoming objectives of your research agenda, and how do you intend to further the discipline by expanding upon the discoveries made in this study?

Reviewer 2 Report

Comments and Suggestions for Authors

Bian et al, performed a study on the spatial and morphological characterstics that may factor into prognosis determination in melanoma. There are several concerns. 

1. There is a lack of validation therefore the issue of overfitting cannot be addressed under the current study design. 

2. The immune cell compartment is also very important. Please explain why only melanoma-focused given that we are in the immunotherapy era. 

3. I am not comfortable with Cox PH model analysis with no adjustment for tumor stage, treatment modality, etc. Those are important factors that have a lot of weight on prognosis. 

Reviewer 3 Report

Comments and Suggestions for Authors

Manuscript Title: Integrating Spatial and Morphological Characteristics into Melanoma Prognosis: A Computational Approach

General Evaluation: This study proposes a computational approach that integrates cellular morphology and spatial configuration features to enhance prognosis analysis of melanoma. The overall research process involves quantifying nuclear size using machine learning and deep learning techniques, followed by assessing its prognostic significance using Cox models. This research direction holds potential clinical utility. However, there are issues with incomplete data and problems with the format of the paper.

Specific Comments:

1. I have noticed that your research only includes data from 26 patients. Is such a sample size sufficient for robust statistical analysis? It is suggested that the authors discuss the potential impact of the sample size on the study's results, or consider conducting external validation.

2. I suggest that the article provides more details regarding the selection, training, and validation of machine learning and deep learning models. For instance, the parameter settings of the models, the cross-validation strategy, and the evaluation metrics for model performance. In addition, the rationale for the selection of univariate and multivariate Cox models, and a discussion on possible statistical biases.

3. In your Discussion section, you mentioned, 'Moreover, the multivariate Cox models demonstrated an improved mean concordance index upon incorporating nuclei size data, indicating a more accurate prognostic performance when including detailed morphological information.' However, it seems that you did not discuss the prognostic performance without incorporating nuclei size data. It is recommended to include this information.

4. Some formatting errors: In Figure 4, 'Ki67+ (red)' should be 'Ki67+ (green)'; a caption should be added to Figure 5.

Reviewer 4 Report

Comments and Suggestions for Authors

This is an interesting study, which examined the prognostic value of cellular morphology and spatial configurations in melanoma, and the results showed the potential of integrating spatial and morphological analyses into melanoma prognostication. However the reviewer has the following concerns.

(1) Can you add one simple summary?

(2) Please avoid using first-person pronouns when writing.

(3) It will be better if the the paragraph from line 47~57 was moved to result or discussion part instead of introduction part.

(4) Figure 5 does not have a caption displayed.

(5) In the discussion part, can you discuss and compare with other studies? Or what’s the difference from others?

(6) In line 81, why was the detection threshold 0.5 chosen , any references?

Comments on the Quality of English Language

The paper is well organized and easy to read, and the writing is clear, concise, and grammatically correct.

Round 2

Reviewer 3 Report

Comments and Suggestions for Authors

The study can be accepted at this version.

Comments on the Quality of English Language

The authors answered all questions that I referred to.